# Risk Riding Behaviors of Urban E-Bikes: A Literature Review

**DOI:** 10.3390/ijerph16132308

**Published:** 2019-06-28

**Authors:** Changxi Ma, Dong Yang, Jibiao Zhou, Zhongxiang Feng, Quan Yuan

**Affiliations:** 1School of Traffic and Transportation, Lanzhou Jiaotong University, Lanzhou 730070, China; 2School of Civil and Transportation Engineering, Ningbo University of Technology, Ningbo 315211, China; 3School of Automotive and Traffic Engineering, Hefei University of Technology, Hefei 230009, China; 4State Key Laboratory of Automotive Safety and Energy, Tsinghua University, Beijing 100084, China

**Keywords:** traffic engineering, e-bikes, risky riding behavior, traffic accidents, interventions

## Abstract

In order to clearly understand the risky riding behaviors of electric bicycles (e-bikes) and analyze the riding characteristics, we review the research results of the e-bike risky riding behavior from three aspects: the characteristics and causes of e-bike accidents, the characteristics of users’ traffic behavior, and the prevention and intervention of traffic accidents. The analysis results show that the existing research methods on risky riding behavior of e-bikes mainly involve questionnaire survey methods, structural equation models, and binary probability models. The illegal occupation of motor vehicle lanes, over-speed cycling, red-light running, and illegal manned and reverse cycling are the main risky riding behaviors seen with e-bikes. Due to the difference in physiological and psychological characteristics such as gender, age, audiovisual ability, responsiveness, patience when waiting for a red light, congregation, etc., there are differences in risky cycling behaviors of different users. Accident prevention measures, such as uniform registration of licenses, the implementation of quasi-drive systems, improvements of the riding environment, enhancements of safety awareness and training, are considered effective measures for preventing e-bike accidents and protecting the traffic safety of users. Finally, in view of the shortcomings of the current research, the authors point out three research directions that can be further explored in the future. The strong association rules between risky riding behavior and traffic accidents should be explored using big data analysis. The relationships between risk awareness, risky cycling, and traffic accidents should be studied using the scales of risk perception, risk attitude, and risk tolerance. In a variety of complex mixed scenes, the risk degree, coupling characteristics, interventions, and the coupling effects of various combination intervention measures of e-bike riding behaviors should be researched using coupling theory in the future.

## 1. Introduction

In recent years, electric bicycles (e-bikes) became the best choice for daily travel of some residents in large and medium-sized cities in China due to their low price, convenience, and flexibility. Unlike in North America and Europe, e-bikes are the main traffic mode in many of China’s major cities and are used primarily for commuting rather than simply for leisure. According to the statistics of the Chinese cycling association [1], in 2017, the total number of e-bikes in China was 250 million, the output of e-bikes was 30.97 million, and the export was 7.301 million, with an export value of $1.44 billion. In Nanning, Haikou, Kunming, Guilin, and other cities, e-bikes far outnumber bicycles; Nanning has more than 1.8 million e-bikes [2], which is known as “the city of electric bicycles”, due to it having the largest number of e-bikes in the country. Therefore, e-bikes are currently one of the most important means of commuter transportation [3].

Despite the obvious advantages, the rapid growth of e-bikes also causes a series of safety problems. Like traditional bicycles and pedestrians, e-bikes also belong to the category of vulnerable groups on the road. Due to their fast speed, e-bikes have more serious accident risks. According to the statistical annual report of China’s road traffic accidents in 2015, the number of e-bike accidents was 8.2 times larger than that of bicycle accidents and 5.4 times larger than that of pedestrian accidents [4]. From January to June 2016, the number of e-bike accidents accounted for nearly 70% of the total number of accidents in Jiangsu Province [5]. The hospital data are not optimistic either. The hospitalization records of e-bike users in Hefei from 2009 to 2011 show that one-third of e-bike users were seriously injured [6]. According to the hospitalization records of Suzhou from October 2010 to April 2011, the number of injured e-bike users accounted for 57.2% of the rate of road traffic hospitalization [7]. In addition to the severity of accidents, the number of e-bike accidents also shows a trend of continuous growth. According to the statistical data [8], the numbers of e-bike traffic accidents in 2011 and 2016 were 10,347 and 17,747, respectively, and the number of deaths increased by 71.52% in the five years. The numbers of e-bike injuries were 11,381 and 19,678, respectively, increasing by 72.90%. Based on the cases of injuries or casualties of two-wheeled vehicles in five cities in China from July 2011–June 2016, electric bicycle accidents were a common type of two-wheeled vehicle accidents in China, accounting for 34.79% of the total number. Among these accidents involving electric bicycles, those causing minor injuries to the riders accounted for 70.0%, while the proportion of serious injuries was 12.6%, and the proportion of deaths was 10.6% [9]. Due to the frequent occurrence and severity of e-bike accidents, cities like Guangzhou, Shenzhen, Wenzhou, and Fuzhou banned or restricted the use of e-bikes [6]. At the same time, in terms of national laws and regulations, relevant provisions were also formulated to restrain illegal behaviors. For example, Article 70 of the regulations for the implementation of the road traffic safety law of the People’s Republic of China stipulates that “when riding a bicycle, an electric bicycle, or a tricycle and crossing a motor vehicle lane on a road, the rider should get off the vehicle and carry it. If there is no crosswalk or pedestrian crossing facilities, or if it is inconvenient to use them, the rider should go straight through after confirming safety.”

Numerous studies showed that human factors, especially the behavior of e-bike users (Figure 1), are important in most traffic accidents, as is the case with e-bikes. Figure 1 shows the keyword co-occurrence network of e-bike safety studies, in which we found that previous studies mainly focused on behavior, safety, risk, crashes, choice, and so on. E-bike traffic violations mainly involve a violation of traffic signals, a violation of regulations on manned vehicles, a failure to drive in non-motorized lanes, adverse driving, etc. [10]. For example, Zhang et al. [11] found that the violation of traffic signals by e-bikes is one of the main causes of road traffic accidents, accounting for 54.18% of the total accident data. Cherry and Du et al. [12,13] analyzed the violations of e-bike users and found that red-light running, over-speeding, and overloading were the main causes of road traffic accidents. Demographic variables, social cognition, and other factors of e-bike users are also closely related to the occurrence of traffic accidents. Hu et al. [6] analyzed the influencing factors of e-bike accidents and found that age, gender, and vehicle type had a significant impact. Yao and Wu [14] established the relationship between safety attitude, risk perception, and violations of e-bike users, and the results showed that both gender and driving experience had a significant impact on e-bike accidents. Papoutsi et al. [15] analyzed the age, gender, accident time, and accident cause of e-bike users using the accident data of a hospital in Switzerland. Guo et al. [16] found that e-bikes are more likely to be involved in red-light running than ordinary bikes. Zhou et al. [17] empirically analyzed the significant factors affecting accidents involving e-bikes and the use of license plates, and found that the use of license plates on e-bikes reduced the possibility of accidents. Guo et al. [18] found from a model comparison analysis that the cyclist’s gender, cycling behavior, type of e-bike, speed limit, and other factors had a significant impact on the severity of e-bike collisions. Based on the previous studies, the visualization of risky riding behaviors of e-bikes can be found through collaboration network analysis, as shown in Figure 2. In Figure 2, each node represents an author or an organization, and the node sizes indicate the number of published papers. The links between nodes represent the collaborations, whereby the greater width of a link represents a closer collaboration. Specifically, the citation network among productive authors is shown in Figure 2a, the co-authorship network among productive authors is shown in Figure 2b, and the collaboration network among research institutions is shown in Figure 2c.

With the increasing attention paid to the problems discussed above, it is urgent to better understand the risky riding behavior of e-bikes and the physiological and psychological characteristics of e-bike users, so as to reduce the accident rate and improve the safety awareness of e-bike users. At the same time, on the basis of studies on the risky riding behavior of e-bikes, it is of great significance to improve road traffic safety and reduce traffic accidents by further analyzing the traffic accident characteristics and traffic accident causes of e-bikes.

## 2. Analysis of Risky Riding Behavior Characteristics and Influencing Factors of E-Bikes

### 2.1. Data Acquisition and Analysis Methods

1. In terms of data collection of risky riding behavior of e-bikes, questionnaire survey methods [19,20,21,22,23,24,25] and video collection methods [26,27,28] are mainly adopted. The questionnaire in the questionnaire survey method was designed based on previous studies on the driving behavior of mopeds, motorcycle users, and automobile drivers. Most of the used items were selected from the moped user behavior questionnaire designed by Steg et al. [19], the motorcycle driver behavior questionnaire designed by Elliott et al. [20], and the Chinese cycling behavior questionnaire designed by Xie and Parker [21]. In the questionnaire design, the items which applied to e-bike users or which could be modified were retained, while the rest were discarded, and new features of e-bike riding were added. Respondents were asked to rate the frequency of each ride behavior on a five-point Likert scale, ranging from “never” (1) to “almost always” (5). Several typical e-bike users were pre-tested, and the questionnaire was revised according to their feedback to improve the clarity and readability.

The questionnaire survey approach is widely used in traffic safety research to gather information such as driving behavior, safety attitude, and risk perception [17,19,20,21,22,23,24]. For example, Yao and Wu [14] studied the risk factors affecting the participation of e-bike users in accidents based on the questionnaire survey method, and determined the relationship between safety attitude, risk perception, and abnormal riding behavior. The survey included 603 e-bike users in Beijing and Hangzhou. The results showed that gender and driving experience were significantly related to traffic accidents. Men were more likely to have accidents than women, and cyclists with a driver’s license were less likely to have accidents than those without one. The study also found that two types of abnormal cycling behaviors, errors and aggressive behavior, were important factors in predicting traffic accidents. In order to study the mechanism and cause of electric bicycle collisions, Hertach et al. [29] used logistic regression model to analyze the questionnaire data of 3659 Swiss e-bike riders. It was found that about 17% of electric bike riders had traffic accidents. The types of accidents mostly involved slipping, falling when crossing the threshold, and slipping when entering a tram/railway. The causes of accidents included the road being too slippery, the riding speed being too fast, a loss of balance while riding, etc. In order to understand the awareness, cycling behavior, and legislative attitude of e-bike riders and non e-bike road cyclists in Tianjin, Wang et al. [30] conducted comparative analysis and research on the two types of cyclists through a large number of questionnaire surveys and interviews. The results showed that 74.2% of e-bike riders thought it was necessary to wear helmets, and 54.7% of e-bike riders thought windshield installation was wrong, which was higher than other road users (49.1% and 48.4%, respectively). However, in field surveys, e-bike rider awareness of various violations lagged far behind what is right. Guo et al. [31] investigated and interviewed 884 e-bike riders in Nanjing to study the influencing factors of e-bike registration in China. The results showed that registration was influenced by gender, age, education, driving license, family car ownership, family income, and electric bicycle travel frequency.

Reason et al. [25] proposed a logical framework to evaluate abnormal driving behavior, and designed the driving behavior questionnaire (DBQ) to distinguish three types of behavior: error behavior (planned action failed to achieve expected result), failure behavior (action intention deviation), and irregularity behavior (deliberately deviating from the normal safety behavior or a socially accepted code of conduct for violations). A modified version of DBQ was also used to study the unusual behavior of two-wheeled vehicles users, such as motorcycle users and moped users. Elliott et al. [20] developed a motorbike driver behavior questionnaire and found differences in traffic errors, control errors, speed violations, stunts, and the use of motorcycle safety equipment in the United Kingdom (UK). Steg and Brussel [19] developed a questionnaire on the behavior of moped users and verified the difference between errors, faults, and violations of moped users in the Netherlands.

The video collection method involves the use of electronic monitoring equipment on the road to observe and collect statistics on the riding behavior and user characteristics of e-bike users, resulting in a relatively rich amount of data collection. Zhou et al. [10] used the “global eye” network video monitoring technology of China telecom to obtain real-time video data of e-bikes in Ningbo, analyzed the main factors affecting the endurance time of e-bikes, and concluded that the weather, crossroad, and the presence of traffic police had the largest influence. Du [13] observed 18,000 e-bike users at the intersection of Suzhou and summarized risky riding behaviors. Konstantina [26] observed 90,000 e-bike users at six monitoring points in Iowa, and studied the influence of road conditions, geographical location, weather conditions, and other factors on the use of helmets. Truing [27] observed 26,000 users of motorcycles and e-bikes, and obtained a correlation between the use of mobile phones while riding and factors such as vehicle type and age. Huan et al. [28] used video monitoring data of intersections to build a model to analyze the influence factors of waiting time of e-bike users and red-light running behavior at intersections, and found that a lower number of e-bike users or a higher number of motor vehicles at an intersection resulted in a lesser likelihood of users exhibiting red-light running behavior.

2. Structural equation models (SEMs) and bivariate probit (BP) models are mainly adopted in the analysis of risky riding behavior data of e-bikes. Before the construction of an SEM or BP model, the data from questionnaire surveys should be tested.

The process of data inspection was composed of three parts. Firstly, exploratory and confirmatory factor analysis was performed to examine the basic dimensions and structures used to detect abnormal cycling behavior in the questionnaire design. Several fitting indexes are usually used, including approximate root-mean-square error (RMSEA), goodness-of-fit index (GFI), adjusted goodness-of-fit index (AGFI), and comparative fitting index (CFI). When the fitting degree of the model is checked, it is required that the RMSEA is lower than 0.08 and the GFI is greater than 0.90, while the AGFI and CFI should indicate a good match between the model and data [32]. Cronbach’s alpha is a consistency coefficient used to measure the homogeneity of items in a single dimension, that is, to evaluate the reliability and internal consistency of scales describing abnormal cycling behavior, safety attitude, and risk perception. According to Nunnally’s standard [33], a value of equal to or greater than 0.7 indicates acceptable reliability.

Secondly, correlations and differences were analyzed. A cycling behavior scale was established to check whether the respondents involved in traffic accidents in the report were significantly different based on demographic variables, risk perception, safety attitude, and abnormal cycling behavior. For univariate analysis, the chi-square test was used for classification variables, and univariate analysis of variance was used for continuous variables. For multivariate analysis, a binary logistic regression model was used to identify factors significantly associated with participants involved in the accident.

Thirdly, the SEM model was built to explore the causal relationship between safety attitude, risk perception, and abnormal cycling behavior using AMOS 17.0 software (SPSS Inc, Chicago, IL, U.S.A.). The two-step program for the SEM model was recommended by Anderson and Gerbing [34]. Firstly, a confirmatory factor analysis was used to evaluate the SEM model and the measurement models of safety attitude, risk perception, and abnormal riding behavior subscales, as well as their fitting degree with their respective potential structures. Secondly, the statistical acceptability of the SEM model was tested, and the maximum-likelihood function estimation model was adopted. The commonly used fitting indexes for inspection [12] of RMSEA, GFI, AGFI, and CFI were used for the fitting of the measurement model.

Several results emerged in the literature based on the above operation steps. For example, based on the SEM model, Yao and Wu [14] found that both safety attitude and risk perception significantly affected abnormal riding behaviors of e-bikes. Guo et al. [35] designed a bivariate probability (BP) model to check the important factors related to the collision of e-bikes and the license plates of e-bikes, and considered the correlation between them. The marginal effect of contribution factors was calculated to quantify their effect on the results. The results showed that gender, age, education level, driving license, family car, experience of using e-bikes, compliance with law, and active driving behavior of e-bike users had a significant influence on e-bike accidents and license plate use. In addition, the type of e-bike, frequency of e-bike use, impulsive behavior, degree of riding experience, and risk perception scale were found to be related to collisions involving e-bikes. This further confirms the previous research conclusion of Yao and Wu [14] that the probability of traffic accidents of e-bike users with a high risk perception is relatively low.

### 2.2. Characteristics of Risky Riding Behavior

E-bike riding is affected by various risk factors, which include users, vehicles, roads and the environment, management, and other aspects [36], as shown in Table 1.

E-bike users have many risky riding behaviors in the process of riding. For example, Brian et al. [38] studied the riding behavior of e-bikes and found that the violation rate of e-bikes was relatively high, and the risky cycling behaviors that caused the violation mainly included not riding in the right direction, over-speeding, traffic conflicts with other road participants, and waiting for the signal light at the intersection. Zhao Ming et al. [39] conducted a four-day traffic survey in Jinhua to study the risky riding behaviors of e-bike users. The results showed that over-speeding, manned riding, red-light running, and driving in reverse were the main risky riding behaviors of e-bike users. Du et al. [13] researched electric bicycle riding behavior and found that there were some risky riding behaviors such as carrying passengers during the riding, taking up lanes, driving through a red light, reverse cycling, and making phone calls. They also found that the number of male users was proportional to the use of helmets and taking up motor vehicle lanes, while the e-bike having a plate was closely related to the behavior of carrying a person or cargo. Miao et al. [40] studied non-motor vehicle users’ risky behavior and found that non-motor vehicle users mainly had behaviors such as occupying the lanes [41], riding side by side in non-motor vehicle lanes, being close to large vehicles riding on the right side, forcing in front of large vehicles when turning right, weaving, drunk riding, over-speeding, and riding in a closed road. In addition, non-motor vehicles also generate a series of risky riding behaviors due to the “exceeding standard” of non-motor vehicles and the wet and slippery road surface through intersections or highway exits on rainy and snowy days. Wang [42], Jia [43], and Jiang and Li [44] found through investigation that cyclists’ risky riding behaviors in the process of cycling included speeding, illegal turning, reverse driving, encroaching on motor vehicle lanes and sidewalks, running red lights, forced overtaking, sudden parking or turning, and non-compliance with regulations.

On the basis of statistical analysis of accident data and causes, Ren [37] gave 12 kinds of risky riding behaviors, and the score of unsafe behaviors is shown in Figure 3. As can be seen from Figure 3, a lower score of unsafe behaviors of cyclists is correlated with a higher frequency of those behaviors, indicating that the traffic accidents and safety hazards of such behaviors are greater. It can be found from Figure 1 that the two unsafe cycling behaviors, manned riding and illegal occupation of motor vehicle lanes, have the highest probability of traffic accidents.

There is a conflict between e-bikes and other road participants at intersections. Many scholars studied the waiting behavior of cyclists at intersections. Guo et al. [15] analyzed the red-light riding behavior of non-motor vehicles at signalized intersections and found that e-bike users were more likely to run the red light than ordinary bikes. Xu et al. [45], in studying the characteristics of the illegal behaviors of mopeds at intersections, pointed out that the illegal behaviors included red-light running, driving in the opposite direction, occupying the motorcycle lane, illegal waiting, and illegal manning or objects. Liu [46], Zhao [47], and Huan Mei et al. [48,49] concluded from previous studies that running a red light was the most serious risky cycling behavior at signalized intersections. Yang et al. [50] studied the traffic light waiting behavior of traditional bicycle users and electric bicycle users at intersections, and found that the probability of cyclists running a red light was proportional to the waiting time, that is, when the red-light time was less than 49 seconds, about 50% of cyclists ran the red light, while, when the time of the red light was less than 97 seconds, about 75% of users ran the red light. They also found that e-bike users were more sensitive to changes in the external environment. Brian et al. [38], Huan et al. [49], and Liu and Yang [51] found that the longer a cyclist waited at an intersection, the higher the violation rate was, which was closely related to herd mentality. In their sample, more than 50% of users could not tolerate 49 seconds or longer, while 25% of cyclists could tolerate 97 seconds or longer. Zhou et al. [10] studied the influencing factors of endurance time at intersections on e-bikes by obtaining 57,213 samples of endurance time of e-bike crossings in Ningbo. Statistics showed that there were three significant variables affecting the endurance time: weather, crosswalk length, and the presence of traffic police to enforce the law. Dong et al. [52] analyzed the safety influence mechanism of green flashing lights on the stopping decision-making behavior of e-bike users, and found that green flashing lights could prompt users to stop, while it was easy to generate aggressive passing behavior. If the green flashing light is set properly, it can effectively improve the safe operation of e-bikes at intersections.

Compared with previous studies, we also found that some factors such as the weather, temperature, and road infrastructure were also closely related to e-bike riding behavior. For example, Konstantina [26] studied the factors affecting helmets worn by e-bike users, and found that the proportion of helmets worn by cyclists on main roads and secondary roads was lower. Du et al. [13] found that, compared with sunny and cloudy days, cyclists wore more helmets in rainy days. In recent years, due to the hot summer weather, the ultraviolet intensity is greater, and the phenomenon of red-light running is more serious. In view of the current situation of weather and temperature, the management department set up sunshades at urban intersections and achieved good results. Zhang et al. [53] studied the cycling behaviors of cyclists at sunshade intersections, and the results showed that the behavior of running red lights at sunshade intersections on sunny days was significantly lower than that on cloudy days. Cyclists at intersections without awnings were significantly more likely to cross red lights. Therefore, the safety of cyclists at intersections can be improved by setting shade awnings at intersections. For example, Zhang et al. [54] explored the violation behavior of non-motor vehicles occupying the motorway, and a binary logistic regression model was adopted to find the internal reason of such violation behavior. The results showed that the traffic violation rate of male cyclists was higher than that of female cyclists, and the violation rate of rainy days was higher than that of sunny and cloudy days. Secondly, the violation rate in the morning peak was higher than that in the evening peak and off-peak hours. The traffic density of motor vehicles and non-motor vehicles had a strong influence on the illegal traffic behavior of non-motor vehicles. Thirdly, the data analysis showed that the average illegal traffic rate of non-motor vehicles was 36.1%, indicating that more than one-third of non-motor vehicles had traffic violations. Wang et al. [55] used the accident data of electric bicycle and motor vehicle collisions as the research object to study the factors affecting the degree of injury of electric bicycle riders. It was found that the factors associated with electric bike rider collisions were violation of signal controls, not according to stipulations to give way, age greater than 60 years old, and male. Among these factors, the ratios of accidents that involved a violation of signal controls and not according to stipulations to give way were 2.201 and 1.495, respectively, indicating that the risk of death or serious injury of electric bicycle riders was greater due to accidents between electric bicycle riders and motor vehicles. The accident occurrence ratios of riders with ages greater than 60 years and male riders were 1.383 and 1.317, indicating that, if the age of an electric bike rider was greater than 60 years, the risk of death or serious injury was higher than that of younger drivers in the event of an accident. This was related to the weaker ability of the older rider to respond to the outside world, in addition to their poor physical condition and the existence of luck. Tang et al. [56] proposed to use a cellular automaton model to study the straight, lane-change, and retrograde behavior of e-bike riders at signalized intersections. The results showed that the lane change and retrograde behaviors had the most significant influence on the riding trajectory of e-bike riders. The simulation results could better explain the complicated traffic phenomenon caused by e-bicycles at signalized intersections. The basis for the introduction of e-bike control measures was provided. Yu et al. [57] studied the reaction of e-bike riders to pedestrian countdown signal devices (PCSDs), and found that PCSDs could effectively reduce the number of red-light violations by cyclists, effectively preventing the running of red lights, but increasing the possibility of early behavior. Fishman and Cherry [58] gave a literature review of the major trends in the development of global electric bicycle traffic from 2006 to 2015. The results showed that, in terms of road safety issues, the violations of Chinese electric bicycle riders at intersections were more common, whereas the risk of electric bicycle accidents was higher than that of traditional bicycles.

In addition, there are some other riding behaviors which can also become safety hazards in traffic accidents. For example, Stelling et al. [59] studied the auditory perception during collisions and accidents of cyclists when they listened to music or made phone calls [41]. The results showed that listening to music and talking on the phone had negative effects on cyclists’ hearing and attention, and there was a greater risk of traffic accidents. They also found that there was no relationship between the frequency of listening to music and making phone calls and the frequency of traffic accidents. Truong et al. [27] investigated and analyzed the mobile phone use of e-bike users, and found that 8.4% of them used mobile phones during riding, and the use of mobile phones during riding was related to riding weather, the number of lanes, different lanes, the duration of red lights, and the presence of police. In addition, wearing safety helmets and some risky riding behaviors at the exit of rain/snow expressways or highways were closely related to the occurrence and severity of e-bike accidents.

## 3. Analysis of Characteristics of E-Bike Users

### 3.1. Vision and Hearing

Vision and hearing are fundamental to riding an e-bike, and they are directly related to the perception of the external environment. The level of perception has a significant impact on the safety of the road. Drunk cycling [25] reduces the visual and other sensory functions of cyclists, and the acquisition and judgment of external information are prone to errors. In addition, the mood of the drunk user is very unstable, and risky riding behaviors such as over-speeding or running a red light are easily generated.

E-bike users have differing levels of dynamic and static vision [41], based on factors such as age and physical feature. The user’s dynamic observation varies according to the speed of cycling, whereby a faster speed results in a narrower dynamic field of vision, while illusions can also occur due to differences between senses, which are important factors in causing traffic accidents. Due to its small size and low driving noise on the road, e-bikes are not easily heard or perceived by other road participants [12]; thus, there are many unsafe and uncertain factors when considering non-motor vehicle lanes and other participants. Zhao [60] studied the visual behavior of e-bike users. Their results showed that, in different cycling environments, the distribution of eye movement time was uneven. A larger proportion of scanning time led to a shorter duration of each fixation point, while a more complex environment narrowed the user’s gaze. In the process of cycling, the following a vehicle can be considered a hazardous situation as electric bikes overtake more frequently.

### 3.2. Different Age Groups

The probability of traffic accidents caused by cyclists of different ages varies due to their wide-ranging physical functions. Wu et al. [61] studied user riding behavior and the relationship with age using survey data, and found that young and middle-aged people were more likely to ride through a red light than older people. They also found that small groups of riders or individual riders were more likely to run red lights, whereas middle-aged and older riders were more fearful of traffic accidents while riding, leading to more cautious behavior [62]. According to the survey data of Truong et al. [27], the average age of e-bike users is 23 years old, while that of motorcycle drivers is 30 years old.

### 3.3. Gender Difference

Users of different genders have different physical skills and psychological states; thus, gender difference also has an impact on the behavior of cyclists. Wu et al. [61] studied the riding behaviors of male and female cyclists through survey data, and found that the probability of male users running a red light was higher than that of female users, especially when the mopeds of male users had greater dynamic performance [41,62,63]. According to the questionnaire survey results of Yao and Wu [13], it was found that gender was significantly related to traffic accidents, whereby men were more likely to have traffic accidents than women. Truong et al. [27] also pointed out that the incidence of mobile phone use in cycling was lower in women than in men.

### 3.4. Reaction Ability

A slow response time and the accuracy of the response are directly related to the occurrence of traffic accidents. Wang [42] summarized the reaction processes of users as follows: external stimulus–conscious acknowledgement–response. In other words, the information obtained through the sensory system is fed back to the e-bike, followed by a certain behavioral operation after the central nervous system makes a decision. The responsiveness is related to the cycling speed of the vehicle and the complexity of the driving environment. For example, the driving speed of an electric bicycle is faster than that of an ordinary bicycle; thus, the requirement for its responsiveness is higher than that of a bicycle user. If the complexity of the driving environment exceeds the processing ability of the cyclists within the allowed range, their reaction ability will not keep up with the changes of the environment, and unsafe riding scenes are easily generated. Fu [64] found that, due to fatigue, drugs, pathology, and other physiological characteristics, cyclists’ consciousness level would decline, their reaction would be relatively slow, and they would be prone to drowsiness and other symptoms. The above symptoms were mainly manifested in cycling behavior with wrong riding and poor judgment.

In summary, different user riding behaviors occur due to several factors, such as gender, age, and the ability to hear and respond to outside stimuli. In cases of drinking, fatigue, illness, and drug use, the brain’s nervous system is in a state of dottiness, which slows reaction time and increases erratic behavior such as running red lights, seriously affecting the user and other participants in terms of road safety.

### 3.5. Psychological Factors

Due to the differences in cyclist psychology, the characteristics of each user are unique. Therefore, on the basis of understanding the general psychology of cyclists, special promotion and education should be carried out to achieve the purpose of safety. Wang [42] pointed out that a normal psychological environment is required for safe cycling. In the process of cycling, people are prone to fear, transcendence, dispersion, conformity, habit, frustration, competitiveness, and distraction, which are hidden dangers leading to traffic accidents. Fu [64] pointed out that, because the speed of e-bikes is faster than that of bicycles, cyclists tend to show exhibit behaviors such as competitiveness, transcendence, conformity, independence, and dispersion, leaving them prone to unsafe riding behaviors (such as speeding, following, running red lights, etc.).

### 3.6. Relationship between Different Factors and Riding Behaviors

It can be seen from the above research that e-bike riders have different personal attributes, and the road traffic infrastructure selected by the riders when riding is different, which will generate different risky riding behaviors. The different variables used and the correlations discussed in previous studies are shown in Table 2.

According to the list of independent variables and dependent variables used in previous studies [10,12,13,25,27,36,37,38,39,40,41,42,43,44,45,46,47,48,49,50,51,52,53,54,55,56,57,58,59,60,61,62,63,64], we found that the characteristics of e-bike users, such as vision, hearing, age group, reaction ability, and psychological factors, are the main reasons affecting the risky riding behavior of cyclists, as well as red light duration and traffic sign marking. For example, Du et al. [13], Wang [42], and Zhang et al. [53] found that independent variables, such as group psychology and waiting too long for the red light, were the main risky riding behaviors related to running a red light. Fu [64] also found that psychological factors, such as transcendence, habit, and distraction, were also the main risky riding behaviors related to over-speed, as well as gender and age. What is more, when cyclists ride on roads with inorganic non-isolation zones, they tend to illegally occupy motor vehicle lanes [55,56].

It can be seen from the above review that the risky riding behaviors of e-bikes are related to the psychological characteristics of users. The psychological characteristics related to safety risks are all caused by the weak subjective safety awareness of cyclists. Therefore, it is expected that, to prevent the occurrence of such risky cycling behaviors, it is necessary to carry out targeted psychological intervention. To sum up, cyclists’ risky riding behaviors are the main factors causing traffic accidents. These behaviors include illegally occupying motor vehicle lanes, running red lights, and illegally carrying people or objects. In general, these risky riding behaviors are the result of a lack of knowledge about the safety features of e-bikes and the weak awareness of road traffic safety. Therefore, it is necessary to strengthen safety awareness and education to improve psychological preparedness. Furthermore, it is necessary to increase cycling skill training to improve coping ability, and to formulate relevant laws and regulations to regulate cycling behavior.

## 4. Prevention and Intervention of E-Bike Traffic Accidents

In view of the prevention and intervention of e-bike traffic accidents, previous research put forward relevant measures based mainly on four aspects: strengthening traffic management, improving laws and regulations, improving the cycling environment of e-bikes, and strengthening the safety education and training of users.
In terms of strengthening traffic management, Ma [65] proposed giving priority to the development of public transportation to limit the increase of the number of e-bikes, following a study of collision experiments between e-bikes and motor vehicles and an analysis of the safety degree of e-bikes. Other suggested measures include limiting the maximum design speed of e-bikes to realize source control, carrying out cycling skill training and assessment, implementing annual inspection, and ensuring the safety of vehicles. Dong [41] proposed that, in order to more clearly identify e-bike riding on the road, reverse laser technology can be adopted on the body and license plate of the e-bike to prevent the occurrence of collision accidents. Jiang and Li [44] proposed to regulate the e-bike industry, strengthen the management of users and e-bikes, strengthen traffic management, improve road conditions, set up e-bike city management departments, and introduce relevant laws and regulations on the basis of analyzing the potential safety risks of e-bikes. Liu and Yang [51] put forward corresponding permit management and compulsory speed limit systems for e-bike users’ red-light running behavior, including (i) training, assessment, and licensing before e-bike riding, and (ii) penalties for illegal riding of e-bikes. Similarly, Shi [66,67] and Chen [68] proposed unified registration and listing for e-bikes, a standardized production of e-bikes, the implementation of a permit system, and e-bike users paying for insurance. Xu et al. [45] emphasized strengthening production management, regulating mopeds, and preventing vehicles from “exceeding the standard”. Wu et al. [69] investigated the status quo of license plate registration of e-bikes, and found that 30.58% of respondents registered their license plates, but there were still 69.42% of users who had unregistered license plates. Carole [70], on the basis of a comprehensive definition of risky riding behavior, suggested that the rapid growth in volume of electric bicycles was consistent with the high electric bicycle accident rate in China, and pointed out that relevant departments should first establish guiding principles for the prevention of traffic accidents through a forward-looking guidance strategy.In terms of improving relevant laws and regulations, Ma [65] advocated that laws should be passed to explicitly prohibit minors from riding e-bikes. Chen [68] and Wu et al. [69] proposed that users must wear hard hats to ride so as to not be fined. Shi et al. [66] and Jang and Li [44] proposed revising and increasing the traffic laws and regulations for e-bikes, forcing cyclists to change their risky riding behaviors through legal means. Truong et al. [27], through an investigation and analysis of mobile phone use of e-bike users, proposed combining existing legislation with extensive education and publicity to reduce potential death and injury caused by mobile phone use during cycling.In terms of improving the cycling environment of e-bikes, Dong [71] and Dong [41] put forward that the separation between machines and non-machines should be ensured. Gu [43] found that speed is the key to controlling user cycling safety factors; thus, measures such as vertical migration (speed humps, deceleration machines, pavement textures, raised crossings, vibrating belts), horizontal deflections (turns), road narrowing, and others (closed road width, central islands, coating surfaces, control of vehicle speed, and deceleration graphics) were suggested aimed at slowing down traffic to reduce hazards faced by electric bicycle users and other road members. Xu et al. [45] proposed improving the transportation infrastructure for e-bikes. Zhao [47] used the time–space isolation method to isolate machines driving directly and turning left at intersections, and set a no-driving area of non-motor vehicles at intersections. They suggested setting left and right turning lanes at intersections, along with special signal lights at intersections with a large number of e-bikes. At small intersections, non-motor vehicles would be required to cross sidewalks, whereby a non-motor vehicle bypass area would be set up at the intersection.In terms of strengthening user safety education and training, Wu [72], Xu et al. [45], Zhao [47], and Chen [68] proposed strengthening safety awareness and education, emphasizing the importance of road safety to consciously improve safety awareness, thereby avoiding risky riding behavior. In addition, Chen [68] pointed out that the quasi-driving mode of motor vehicles should be imitated, and the skills of cyclists should be trained to improve their ability and experience.

In addition, Wang et al. [73] found through investigation that the current phenomenon of children being carried on e-bikes is quite common, and pointed out that the safety guarantee of existing electronic bicycle seats is limited. Thus, the design strength and structure of the seats should be reasonably strengthened to ensure the personal safety of children when riding. Yuan et al. [74] analyzed 150 traffic accidents of e-bikes in Beijing, and found an increase at intersections and in motor vehicle lanes in the suburbs where the occurrence of e-bikes was high. Safety education and training should be carried out for users in the suburbs, and the design of traffic safety facilities in these areas should be strengthened at the same time. Yuan et al. [75], using statistical analysis of vulnerable road users in Beijing collisions, compared pedestrian, bicycle, and electric bicycle accident frequency, severity, and influencing factors. It was concluded that electric bicycle accidents were dependent on collision speed and position of the victim, with similarities and differences compared to the other groups. Suggestions were given for accident intervention.

All in all, scholars put forward comprehensive prevention measures for electric bicycle traffic accidents on the basis of traffic management, laws and regulations, riding conditions, and safety education and training, including uniform registration licenses, a quasi-driving system, improvement of the road and intersection traffic safety facilities, strengthening safety awareness and education, driving skill training, and more. These approaches would help improve the traffic safety consciousness of the user while safeguarding them.

## 5. Research Prospect

As a future research direction of urban road traffic safety, it is of great significance to study cyclists’ risky riding behaviors to reduce the occurrence of traffic accidents involving e-bikes and to ensure the safety of the life and property of traffic participants [76,77,78,79,80,81,82]. Although the current traffic safety problems of e-bikes attracted the attention of scholars, due to the difficulties in obtaining traffic accident-related data and the limitations of research ideas, there are still many deficiencies in the study of e-bike risky riding behaviors, which need to be further studied.
In current electric bicycle risky riding behavior research, most historical data are obtained from traffic accidents or sampling survey results for statistical analysis and research; however, due to incomplete traffic accident data, the analysis results have certain deviation. Instead, studies can make use of advanced monitoring systems and video recognition technology to obtain more comprehensive electric bicycle traffic accident data, with the analysis of big data to mine risk cycling association rules between behavior and traffic accidents, to find a strong relationship between various influencing factors. Using these results, strongly correlated risky riding behaviors can be used to put forward effective intervention measures, in order to effectively cope with frequent electric bicycle traffic accidents [83,84,85,86]. E-bike users also range in age from 18 to 80. Due to the different ages of cyclists, their physical function and cycling behavior are different. Currently, in terms of studies on cyclists’ behaviors at different ages, the only relevant research achievements are based on the behavior of running red lights, while no research was carried out on other risky cycling behaviors, which are in urgent need of study, such as the implementation of an age threshold to ensure the safety of e-bike riding.The high accident rates and casualty rates of e-bikes are mainly caused by risky riding behavior cue to the weak subjective risk awareness of users, and the current research results in this regard are not substantial enough. In view of this, it is possible to strengthen the research on cyclists’ subjective risk awareness in future studies. For example, using scales of risk perception, risk attitude, and the degree of risk tolerance, questionnaire surveys can be conducted on user risk consciousness. Then, correlation analysis and regression analysis can be used to study the relationship between electric bicycle riding behavior and risk awareness, according to different personalities and demographic variables. This study would be helpful to explore the differences in risk awareness among different cyclists. According to these differences, safety education and training can be carried out in a targeted way, so as to ultimately achieve the goal of preventing risky riding behavior.As a commuting tool for some urban residents, e-bikes are often used in rainy, foggy, or snowy days. At present, there are not many research results on risky riding behavior of e-bikes under special weather conditions. In view of this, quantitative research should be undertaken, studying e-bike riding rules in special weather conditions such as rain, fog, or ice and snow. A user safe riding environment threshold should be determined to prioritize electric bicycle safety, thus avoiding risky behavior and reducing traffic accidents. In addition, although many studies looked at traffic safety involving e-bikes, there are few studies on the psychological and physiological characteristics of cyclists. On the one hand, research related to user personality, cognition, self-assessment, and self-adjustment is lacking. On the other hand, research on user attention, analysis ability, spatial ability, emergency ability, and nervous system and physiological characteristics such as cognitive ability is also missing.In the future, it will be interesting to capture the data of the current research on e-bike behavior, as well as the related technologies to be applied. Moreover, by obtaining accurate cycling data, researchers can explore the potential impact of cyclists’ personal characteristics on cycling behavior, and address these issues through safety education and driving training.

## 6. Discussions

E-bike riders in different regions have different risky riding behaviors due to differences in regional culture and traffic regulations. For example, China’s data acquisition for e-bike riders generally includes (a) riding conflicts and illegal riding behaviors, usually observed through video surveillance equipment at intersections [10,13,27,28,58], and (b) accident data of hospitals, violation records, and personal injury records [58]. Research data on behaviors of e-bike riders were also obtained using questionnaires methods [17,19,20,21,22,23,24,29,30,31].

Overall, the risky riding behavior of e-bike riders in China and Europe is similar. In China, previous studies [58] found that the risky riding behaviors included double riding (or carpooling), chatting while riding, listening to music while riding, calling while riding, riding in parallel, over-speeding, jaywalking, reverse riding, suddenly crossing the road, speeding too fast when turning, illegal occupation, failure to comply with regulations, forcibly overtaking, suddenly stopping or turning, and fatigue riding. In Europe, previous studies [26,38] found that the risky riding behaviors included riding in the wrong direction, over-speeding, traffic conflicts with other e-bike riders, jumping the lights, not wearing helmets, listening to music while riding, making phone calls while riding, etc. From the above, the regional issues regarding e-bike behavior showed some differences between China and Europe, since the laws, regulations, and characteristics of e-bike users are different [58].

## 7. Conclusions


Currently, e-bike traffic accidents occur frequently, and more than 90% of e-bike traffic accidents are caused by cyclists’ risky riding behaviors, including illegal occupation of motor vehicle lanes, over-speeding, running red lights, and going against the traffic flow. It is of great significance to study the risky riding behavior of e-bikes, researching accident characteristics so as to prevent them and ensure the safety of people’s life and property.Substantial previous research was carried out on traffic accident characteristics and causes of e-bike risky behavior, as well as their prevention and intervention. However, after reviewing the relevant literature at home and abroad, the authors found that the existing relevant research results have three deficiencies, and pointed out research directions that can be further explored in the future.Real cycling scenes are often a combination of a variety of situations. For example, users of different ages ride electric bikes equipped with umbrellas in rainy or snowy weather. In view of this, it is necessary to study the degree of danger of combination scenarios featuring electric bicycle riding behavior, whereby the degree of danger is not simply a function of the risky riding behavior in a single scene. This research requires quantitative analysis using the theory of coupling characteristics of various scenarios seen in risky riding behavior. On the basis of effective interventions and the coupling effect of various combinations of interventions to achieve an elimination of accident risk, one can ensure traffic safety.


## Figures and Tables

**Figure 1 ijerph-16-02308-f001:**
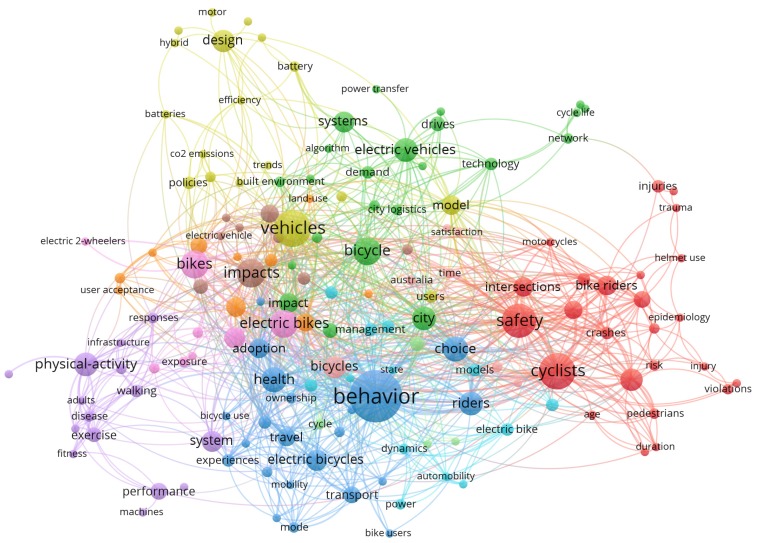
Keyword co-occurrence network of electric bicycle (e-bike) safety studies.

**Figure 2 ijerph-16-02308-f002:**
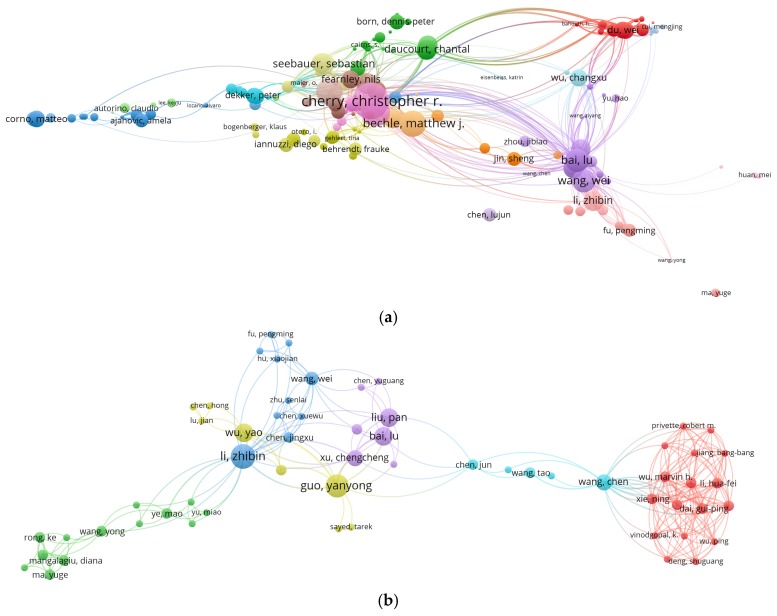
Visualization of risky riding behaviors in e-bike studies: (**a**) the citation network among productive authors; (**b**) the co-authorship network among productive authors; (**c**) the collaboration network among research institutions.

**Figure 3 ijerph-16-02308-f003:**
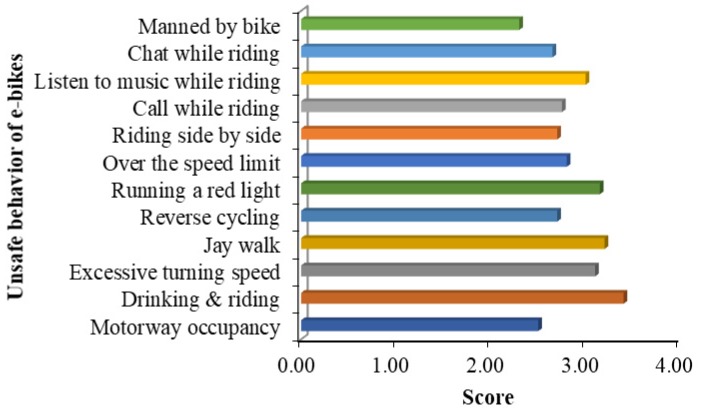
Evaluation of unsafe behavior of e-bikes.

**Table 1 ijerph-16-02308-t001:** Influencing factors of electric bicycle (e-bike) risky riding behavior.

Influencing Factor	Factors Set
Rider factors	Age, gender, education level, health status, personality characteristics, traffic safety awareness, cycling behavior, cycling technology, decision-making ability
Vehicle factors	Braking performance, steering performance, comfort
Road and environmental factors	Traffic flow volume, speed, width of non-motorized lane, form of road section, condition of road surface, conflict interference type, weather conditions, artificial environment
Management factors	Risk management, organization, risk perception, communication
Other factors [37]	Alcohol, drugs, social norms, confidence

**Table 2 ijerph-16-02308-t002:** Independent and dependent variables used in previous studies.

Independent Variables		Dependent Variables
Vision		Manned by bike
Hearing		Chatting while riding
Different age groups		Listening to music while riding
Gender difference		Calling while riding
Reaction ability		Riding side by side
Psychological factors	Fear	Over the speed limit
	Transcendence	Running a red light
	Dispersion	Reverse driving
	Conformity	Jaywalking
	Habit	Excessive turning speed
	Frustration	Drinking and driving
	Competitiveness	Motorway occupancy
	Distraction	Not adhering to stipulations to give way
Aircraft non-isolation belt		Forced overtaking
Red light duration		Sudden stopping or turning
Traffic sign marking		Fatigue riding

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
