# Peer review of "Risk Riding Behaviors of Urban E-Bikes: A Literature Review"

_ijerph, 2019, doi:10.3390/ijerph16132308_

Round 1
Reviewer 1 Report
This manuscript is a review about the risk riding behaviors of urban E-bikes.
It is a good quality manuscript. Some suggestions to improve are:
- Add more research papers-studies related to latest years (2018-2019).
-Add list of the different variables used and correlation discussed in form of table in previous studies.
- Discuss the regional issues (e.g. from different parts of Europe) regarding E-bikes and policy/behaviour and also do comparative analysis.
Author Response
We would like to thank the reviewers for their constructive comments, which have significantly improved the quality of the paper. We have revised the manuscript according to your constructive suggestions, where the corresponding changes to your comments have been highlighted in RED in the revised version of the manuscript. The corresponding responses to your insightful comments are attached in Response-ijerph-514757-Reviewer 1.

Reviewer 2 Report
The paper entitled "Risk Riding Behaviors of Urban E-bikes: A Literature Review" took
the risky behaviors of electric bicycle drivers as the research point. Through a large amount of literature-review, it was found that there were mainly three aspects of research: the characteristics and the causes of the e-bike accidents, the characteristics of users' traffic behavior as well as the prevention and intervention of traffic accidents. The existing research methods on the risky riding behaviors of e-bike mainly involved questionnaire survey method, structural equation model and binary probability model. Then the paper points out three research directions that can be further explored in the future.
On the whole, this is an interesting piece. The literature review at the start is well structured,
and the structure of the article is good. There are a few logical failure, but nothing
that detracts from the aim of the article.
In terms of specific comments, I have a few suggestions:
1.The content in the text does not mention Figure 2, if there is an association, please indicate, if there is no association, you can delete.
2.Can non-motor vehicle terminals acquiring data of the current research obtain E-bikes behavior, and if so, it is recommended to supplement.
3.You had better reorganize the second last paragraph of characteristics of the riding behaviors to strengthen the logical relationships with the above paragraphs.
4.Strengthen the links between Research Prospect and the characteristics of the riding behaviors and user behaviors.
Author Response
We would like to thank the reviewers for their constructive comments, which have significantly improved the quality of the paper. We have revised the manuscript according to your constructive suggestions, where the corresponding changes to your comments have been highlighted in RED in the revised version of the manuscript. The corresponding responses to your insightful comments are attached as Response-ijerph-514757-Reviewer 2.

Reviewer 3 Report
This paper gives a broad overview of the safety of e-bikes. The focus of the paper is on the behavioural aspects. However, the position of the e-bike in the (Chinese) traffic system (importance for the mobility of many people, the presence of infrastructural facilities for bikes and e-bikes) is not quite clear yet. Furthermore, it should be stressed that a traffic crash is always caused by a combination of man-vehicle-road, not only by one of these three factors.
The use of the English language should be checked thoroughly, in particular the use of articles.
The Figures 1 and 2 (a. b, and c)) are interesting. However, these figures are not really treated in the text. The figures should either be moved to an Appendix or be treated in the text..
Page 1, line 21 and Page 6, line 205
What is meant by ‘reverse cycling’? Do you mean cycling in a direction that is prohibited and/or cycling on the wrong side of the road?
Page 2, lines 61 and 62
The number of accidents in 2011 and 2016 was 73 and 1305 respectively while the number of injuries was 8,532 and 16,944 respectively. Are these numbers right? Because in general, the number of accidents should be in the same order of magnitude as the number of injuries.
Page 3, lines 88-90
Please explain why the use of a license plate can reduce the possibility of an accident.
Page 7, Figure 3
The category ‘reverse drinking’ is presumably not right. See also the previous remark about reverse cycling.
Page 12. Line 450-453
Concerning red light running, cyclists sometimes have to wait very long. Perhaps their behaviour can also be influenced positively by shortening the red light phase for cyclists (for instance by providing two green phases for them during one traffic signal cycle.
Page 12, lines 454-465
“… the current research results … are not in-depth enough.” How deep should this be? Is this really necessary for setting up educational programmes? And how about the expected results of these kind of programmes? How effective are these programmes anyway?
Author Response
We would like to thank the reviewers for their constructive comments, which have significantly improved the quality of the paper. We have revised the manuscript according to your constructive suggestions, where the corresponding changes to your comments have been highlighted in RED in the revised version of the manuscript. The corresponding responses to your insightful comments are attached as Response-ijerph-514757-Reviewer 3.
